# OpenReview forum: "InfiMed-ORBIT: Aligning LLMs on Open-Ended Complex Tasks via Rubric-Based Incremental Training"
_ICML.cc/2026/Conference — ICML 2026 regular_

### Official Review · Reviewer_Zs2F · 2026-03-09

**Soundness:** 3
**Presentation:** 3
**Significance:** 3
**Originality:** 2
**Overall Recommendation:** 4
**Confidence:** 3

**Summary:**

This paper addresses the alignment challenge in medical dialogue. Traditional RL lacks reliable and low-cost scalar rewards for such tasks, while reward models are expensive and fragile. The paper proposes ORBIT, which starts from a small set of seed rubrics and combines retrieval-augmented in-context rubric generation, dual difficulty filtering for both samples and rubrics, and incremental reinforcement learning based on GRPO to construct interpretable training signals. Experimental results show significant improvements on HealthBench-Hard, and further validation on InfoBench demonstrates that the framework is not only applicable to the medical domain but may also benefit more general open-ended instruction-following tasks.

**Compliance With Llm Reviewing Policy:**

Affirmed.

**Final Justification:**

I appreciate the authors’ rebuttal, which addressed my main concerns. As a result, I maintain my original score and overall assessment.

**Key Questions For Authors:**

1.The results might be more convincing if the model were re-evaluated using judge models of different scales or from different model families.

2.How can the quality of automatically generated rubrics be quantitatively evaluated?

**Limitations:**

Yes

**Strengths And Weaknesses:**

Strengths:

1.The problem addressed in this work is reasonable and meaningful. The authors clearly point out that open-ended medical dialogue cannot be effectively captured by a single scalar reward. Therefore, they adopt rubrics to decompose multi-dimensional quality criteria and use a judge model to convert whether a criterion is satisfied into more fine-grained rewards. This design aligns well with the characteristics of the task.

2.The overall structure of the paper is clear. The presentation flows smoothly from motivation and method design to experimental setup and ablation studies.

3.The problem studied in this paper is important, particularly in the context of open-ended medical dialogue.

4.The originality of the paper mainly lies in the combination of existing techniques and their adaptation to the task, rather than in proposing a completely new RL algorithm.

Weaknesses:

1.Some experimental hyperparameter settings could be described in greater detail.

2.The paper claims “Proposing ORBIT, a fully automated rubric-based post-training framework for scalable open-ended alignment of LLMs,” but the approach still relies on human-written seed rubrics in practice.

3.The reward signal depends on a single judge model to determine whether rubrics are satisfied, and bias in the judge model may affect the training objective.

---

> ### Author Rebuttal · Authors · 2026-03-31
>
> We thank the reviewer for the constructive review and for recognizing the importance of rubric-based rewards for open-ended medical dialogue.
> We understand the main remaining concerns to be: (i) the phrase ``fully automated'' overstates the current implementation because ORBIT still uses a small expert-written seed pool, (ii) judge dependence should be validated more directly across scales and model families, (iii) rubric quality should be quantified more explicitly, and (iv) some training details should be more clearly reported.
>
> > **Q1**: The approach still relies on human-written seed rubrics in practice.
>
> **A1:**  We agree that our current pipeline relies on a small expert-annotated seed pool, and we will make this point more explicit in the revised Limitations section. However, to explore scenarios in which expert annotations are unavailable, we investigate synthesizing seed rubrics using a multi-LLM ensemble (Gemini-3, DeepSeek-R1, GPT-OSS-120B, and DeepSeek-V3.1), with the outputs summarized by GPT-5.1. By comparing responses across models of varying performance levels, we derived a tailored Multi-LLM-Syn-Contrast strategy. As shown by the results on a 320-example subset below, this zero-seed contrastive variant matches the performance of the Human-RAG baseline. While this demonstrates that our framework can, in principle, be bootstrapped without human annotations, synthesizing these seeds costs more than 80 times as much as our proposed method. Therefore, utilizing a minimal, high-value set of expert-authored seeds remains the most practical and efficient approach for data scaling, and we will include this detailed cost-performance analysis in the revision.
>
> | Variant | Retrieval | Seed Source | Steps | HealthBench-Hard |
> |-|-|-|-|-|
> | Human-RAG (sub) | Yes | Human seed | 140 | 15.34 |
> | Multi-LLM-Syn-Direct (sub) | No | LLM-synthetic | 140 | 14.11 |
> | Multi-LLM-Syn-Contrast (sub) | No | LLM-synthetic + contrastive filtering | 140 | 15.60 |
>
> > **Q2**: The results might be more convincing if the model were re-evaluated using judge models of different scales or from different model families.
>
> **A2:** To address this concern, we directly evaluate judge dependence through a comprehensive cross-judge study. We trained the ORBIT pipeline using judges of varying scales (Qwen3-4B to 30B) and a different model family (GPT-OSS-20B), subsequently evaluating all checkpoints against a fixed, independent evaluator (GPT-OSS-120B) for relative trend analysis.
>
> | Train Judge | Params | Steps | Eval Judge |Avg Score |
> |-|-|-|-|-|
> | Qwen3-4B-Instruct-2507| 4B| 0| GPT-OSS-120B  | 7.22 |
> | +Qwen3-4B-Instruct-2507| 4B| 200| GPT-OSS-120B  | 13.64|
> | +Qwen3-8B | 8B | 200| GPT-OSS-120B| 14.56 |
> | +Qwen3-30B-A3B-Instruct-2507| 30B| 200 | GPT-OSS-120B  | 17.89 |
> | +GPT-OSS-20B| 20B|140| GPT-OSS-120B  | 17.20 |
>
> As the results indicate, although the judge’s capability directly influences the final performance ceiling, even a small baseline model such as Qwen3-4B-Instruct-2507 achieves substantial gains when used for both training and self-judging. More importantly, training with a judge from an entirely different model family (GPT-OSS-20B) yields substantial performance gains even early in training. This provides evidence that ORBIT does not simply overfit to the stylistic bias of a specific model family.
>
> >**Q3**: How can the quality of automatically generated rubrics be quantitatively evaluated?
>
> **A3:** We currently evaluate rubric quality through two quantitative proxies: (1) Downstream utility, as training with high-quality, contrastively filtered rubrics substantially outperforms generic ones (Single-No-RAG) and nearly matches the Human-RAG baseline; and (2) Rubric informativeness, as filtering criteria by estimated pass rates improves training efficiency without degrading final performance. To provide a more direct rubric-level evaluation, we will include in the revision an expert assessment of criterion correctness, specificity, and non-redundancy, alongside a detailed failure analysis.
>
> >**Q4**: Some experimental hyperparameter settings could be described in greater detail.
>
> **A4:** We agree and will comprehensively detail these settings in the revised manuscript and appendix. For the main 2k setting, the key hyperparameters include sampling 8 responses per prompt (temperature 1.0 and top-p 0.95, with validation using top-p 0.7), token-mean loss aggregation, training and generation batch sizes of 128 and 256 with dynamic batching, and filtering over up to 10 generated batches. We will also specify the exact judge checkpoints and the 200-step sweep budget used in each study. Furthermore, we will expand the Limitations section to explicitly discuss the reliance on a small seed pool, sensitivity to judge quality, and importantly, ORBIT's scope as an assistive research framework rather than a fully autonomous clinical system.

---

> > ### Author Rebuttal · Reviewer_Zs2F · 2026-04-01
> >
> > The authors have adequately addressed my main concerns raised in the original review. In particular, their rebuttal clarifies the technical points I was uncertain about and provides sufficient explanations regarding the experimental/theoretical issues. Based on these clarifications, my previous concerns are fully resolved.

---

> > > ### Author Response · Authors · 2026-04-06
> > >
> > > We sincerely thank Reviewer Zs2F for the thoughtful response. The questions and feedback are very helpful to us. Following your suggestions, we will strengthen the evaluation of the generated medical rubrics and provide more details on the experimental hyperparameters in the revised manuscript. We are very grateful for your time and for your valuable comments.

---

### Official Review · Reviewer_GW2c · 2026-03-12

**Soundness:** 4
**Presentation:** 4
**Significance:** 4
**Originality:** 3
**Overall Recommendation:** 4
**Confidence:** 4

**Summary:**

This paper presents ORBIT (open-ended rubric-based incremental training), a rubric-guided reinforcement learning framework for aligning LLMs on open-ended medical dialogue. The method combines synthetic dialogue construction, retrieval-augmented rubric generation from HealthBench seed rubrics, difficulty filtering, and GRPO-based policy optimization to replace opaque scalar reward models with structured rubric-based feedback. Applied to Qwen3-4B-Instruct, ORBIT improves HealthBench-Hard from 7.0 to 27.5 with 2k training samples and scales further with more data, while also showing gains on InfoBench (improving from 42.0 to 82.9 on the hard split).

**Compliance With Llm Reviewing Policy:**

Affirmed.

**Final Justification:**

I thank the authors for the detailed and constructive rebuttal. The additional experiments on out-of-distribution transfer and the zero-seed multi‑LLM synthesis variant help clarify how far the framework can generalize beyond the original HealthBench seeds, and the discussion of contextual prioritization among potentially conflicting rubric criteria makes the reward design more transparent. The clarification about filtering dynamics and learning starvation and the explicit rubric-schema contamination statistics are also helpful, and I appreciate the plan to adopt “dialogue/consultation” terminology in place of “QA” in Section 3.1. Overall, my primary technical concerns have been fully addressed; I continue to view this as a solid and practically useful contribution, and not a conceptually transformative one, so I am keeping my 4/weak accept recommendation.

**Key Questions For Authors:**

1. The framework depends on a seed set of expert-written HealthBench rubrics. What is the minimum viable diversity of this seed set for transfer to a new specialty, language, or patient population without HealthBench-style seeds?
​
2. How does the rubric generator handle conflicts between criteria, such as completeness versus conciseness or reassurance versus escalation, beyond scalar point assignments in the prompt?
​
3. What fraction of training batches is typically removed by the variance-aware dynamic sampling filter, and does this ever create a meaningful gradient starvation issue in practice?
​
4. The contamination analysis is useful, but it focuses mainly on prompt-level leakage. In this setting, should contamination also be analyzed at the rubric-schema level, given that the reward structure is derived from HealthBench seed rubrics?

**Limitations:**

Mostly yes. The authors acknowledge the text-only scope and the dependence on expert rubric seeds. I would encourage a stronger discussion of the clinical deployment gap: improved rubric compliance on consultation benchmarks is important, but it is still far from end-to-end clinical readiness, especially without multimodal grounding or a clearer characterization of performance on rare and high-risk edge cases.

**Strengths And Weaknesses:**

Soundness: The paper’s main strength is that it does more than report a benchmark gain: it studies whether the gain reflects genuine policy improvement rather than sampling luck. The distributional analysis of average rubric compliance and Best-of-N rubric hit rate is especially useful here, because it supports the claim that ORBIT expands the feasible solution space rather than merely exploiting inference-time variance. The contamination controls around HealthBench-Hard and the ablations on filtering and training stability also strengthen the empirical case. The main weakness is that the safety analysis remains incomplete: the paper does not provide a clear failure taxonomy for which negative rubric criteria remain most often violated after RL, which would help characterize residual clinical risk.
​
Presentation: The paper is clearly written and the overall pipeline is easy to follow despite having several moving parts. The figures are informative, especially the ones distinguishing average compliance from Best-of-N effects. One weakness is that the paper understates one of its most important conceptual advantages: it builds an interpretable rubric-based alignment pipeline, but does not fully articulate why auditable rubric criteria are preferable to opaque scalar reward models in high-stakes medical settings where post-hoc safety analysis matters. A minor terminology issue is that the paper sometimes uses “QA” or “Dialogue QA simulation” to describe what is more accurately an open-ended, multi-turn consultation setting (subsection 3.1). Since one of the paper’s central claims is that it goes beyond QA-style optimization, using “dialogue” or “consultation” terminology more consistently would make the framing clearer.

Significance: This is the paper’s strongest dimension. Open-ended medical dialogue is an important and under-solved alignment setting, and the result showing that a 4B model can move from 7.0 to 27.5 on HealthBench-Hard with only 2k samples is practically meaningful. The paper’s main strength is the quality of execution: the empirical gains are strong, the benchmark setting is meaningful, and the additional analysis helps support the claim that the improvements reflect genuine policy change rather than inference-time sampling effects. Its main limitation is that the contribution is primarily a well-executed training framework rather than a fundamentally new theoretical idea.

Originality: The components themselves are mostly established, including GRPO, retrieval-augmented prompting, LLM-as-judge evaluation, and synthetic dialogue construction. The novelty is in how these pieces are combined into a coherent rubric-based RL pipeline for ambiguous, non-verifiable medical dialogue, together with the stability mechanisms and the analysis showing that alignment is not reducible to simple inference scaling. Under ICML’s broader notion of originality, this is a meaningful contribution.

---

> ### Author Rebuttal · Authors · 2026-03-31
>
> We thank the reviewer for the careful and encouraging feedback.
> We especially appreciate the distinction between true policy improvement and mere inference-time sampling effects, as well as the emphasis on the auditability of rubric-based alignment in a high-stakes setting.
>
> > **Q1**: The framework depends on a seed set of expert-written HealthBench rubrics. What is the minimum viable diversity of this seed set for transfer to a new specialty, language, or patient population without HealthBench-style seeds?
>
> **A1:** To address the framework's transferability and its reliance on the seed set, we evaluate our approach from two perspectives: (1) Out-of-Distribution (OOD) generalization with existing seeds, and (2) Zero-seed adaptation for entirely new domains.
>
> 1. Out-of-Distribution (OOD) Transferability: First, our framework demonstrates strong generalization to new domains without requiring new seed rubrics. We evaluated our ORBIT-Med (2k) model on a completely unseen open-ended medical benchmark [1]. On 200 cases, our pipeline consistently improves over the baseline, supporting the robustness and viability of the existing seed set.
>
> | Variant | Task Accuracy| Task Robustness | Info Coverage |Info Relevance|
> |-|-|-|-|-|
> | Qwen3-4B-Instruct-2507| 53.5 | 79.5 | 24.7| 86.3|
> | ORBIT-Med(2k) | 58.0 | 78.9| 26.2| 87.1|
>
> 2. Transferring to Zero-Seed Scenarios (Multi-LLM Synthesis):
> For entirely novel medical settings in which expert-authored seed rubrics are unavailable, we explore a zero-seed alternative: synthesizing a rubric pool using a multi-LLM ensemble. We test two variants on a 320-example subset:
> - Multi-LLM-Direct: Generating and summarizing rubrics using multiple advanced LLMs.
> - Multi-LLM-Contrast: Deriving rubrics by contrasting responses from models of varying performance levels.
>
> | Variant | Retrieval | Seed Source | Steps | HealthBench-Hard|
> |-|-|-|-|-|
> | Human-RAG (sub)|Yes | Human seed |140|15.34|
> | Multi-LLM-Syn-Direct (sub) | No | LLM-synthetic|140|14.11|
> | Multi-LLM-Syn-Contrast (sub) | No | LLM-synthetic + contrastive filtering | 140 | 15.60 |
>
> **Conclusion & Cost Trade-off**: As shown above, the Multi-LLM-Contrast approach achieves performance comparable to our Human-RAG baseline without requiring any human seeds. However, synthesizing these seeds costs >80x more than our proposed method. Therefore, while our ORBIT framework can effectively adapt to new scenarios with zero human seeds via LLM synthesis, utilizing a minimal, high-value set of expert seeds remains the most efficient and scalable approach. We will include this detailed analysis and the multi-LLM synthesis pipeline in the revised paper.
>
> [1] The Dialogue That Heals: A Comprehensive Evaluation of Doctor Agents' Inquiry Capability. arXiv:2509.24958.
>
> > **Q2**: How does the rubric generator handle conflicts between criteria, such as completeness versus conciseness or reassurance versus escalation, beyond scalar point assignments in the prompt?
>
> **A2:** We agree with the reviewer's insight. Rather than relying on an explicit symbolic conflict-resolution module, ORBIT handles criteria tensions through contextual prioritization. Specifically, the rubric generator is conditioned on the current consultation and retrieved examples to produce case-specific, non-redundant criteria, avoiding a single global notion of "goodness." The judge then evaluates these criteria individually, ensuring a structured, multi-dimensional reward instead of a simple scalar. In practice, this allows trade-offs to naturally adapt to the context: escalation and safety-netting dominate in red-flag cases, while conciseness is rewarded in low-risk scenarios provided critical safety constraints are met. We will clarify this mechanism as contextual prioritization, rather than formal constraint satisfaction, in the revision.
>
> > **Q3**: How much filtering occurs, and does it starve learning?
>
> **A3:**  While filtering does lead to some learning starvation in later stages (after nearly 600 steps; see Fig. 4), we actively mitigate this effect. We employ a dynamic batch size (size=10) for gradient accumulation and a multi-stage training strategy (restart + temperature adjustment) to expand exploration, which effectively prevents starvation and yields further gains (Table 2). The exact filtering and effective update ratios will be detailed in the revision.
>
> > **Q4**: The contamination analysis of the rubric-schema level is useful.
>
> **A4:** We agree that rubric-only overlap analysis is crucial. Our analysis shows minimal contamination between Hard and No-Hard sets: exact overlap on example-level rubrics is only 1.08%, >0.90 cosine similarity is just 0.54%, and corpus-level 5-gram Jaccard overlap is 1.56%. This confirms our benchmark gains are not driven by trivial rubric-schema leakage.
>
> > **Q5**: Suggestion to use "dialogue/consultation" instead of "QA" for clearer framing in Section 3.1.
>
> **A5:** We agree with this suggestion and will update it in the revised manuscript.

---

> > ### Author Rebuttal · Reviewer_GW2c · 2026-04-02
> >
> > I thank the authors for the detailed and constructive rebuttal. The additional experiments on out-of-distribution transfer and the zero-seed multi‑LLM synthesis variant help clarify how far the framework can generalize beyond the original HealthBench seeds, and the discussion of contextual prioritization among potentially conflicting rubric criteria makes the reward design more transparent. The clarification about filtering dynamics and learning starvation and the explicit rubric-schema contamination statistics are also helpful, and I appreciate the plan to adopt “dialogue/consultation” terminology in place of “QA” in Section 3.1. Overall, my primary technical concerns have been fully addressed; I continue to view this as a solid and practically useful contribution, and not a conceptually transformative one, so I am keeping my 4/weak accept recommendation.

---

> > > ### Author Response · Authors · 2026-04-06
> > >
> > > We sincerely thank Reviewer GW2c for the thoughtful response. Following your suggestions, we conducted additional experiments about medical seed rubrics. We also added results on a new out-of-distribution medical benchmark. These additions will be incorporated into the revised manuscript. We will also clarify Section 3.1 and discuss some failure cases caused by conflicting rubrics. Thank you again for your constructive and insightful feedback.

---

### Official Review · Reviewer_55vi · 2026-03-12

**Soundness:** 2
**Presentation:** 3
**Significance:** 2
**Originality:** 2
**Overall Recommendation:** 4
**Confidence:** 3

**Summary:**

This paper presents ORBIT, which is a framework to help LLMs do better on open-ended, high-stakes medical conversations. The issue is that regular RL struggles because scalar rewards are vague in these tasks. ORBIT uses rubrics instead, breaking down what makes a good response into smaller, checkable criteria. These rubrics are dynamically retrieved, filtered for difficulty, and then used for incremental reinforcement learning. Testing this on Qwen3-4B-Instruct, the HealthBench-Hard score skyrocketed from 7.0 to 27.5 with only 2k samples. Larger rubrics and more data further improve results, and ORBIT also works on InfoBench. A general aspect discussed by this study is the shift from opaque reward signals to transparent, rubric-based guidance. Overall, the authors outline the concept thoroughly, and it seems like a really clever way to align models without needing huge datasets or massive parameters.

**Compliance With Llm Reviewing Policy:**

Affirmed.

**Final Justification:**

I suggest that this paper be accepted. After carefully assessing the original manuscript's quality as well as the authors' reply, I think this work satisfies publishing requirements and shows adequate contributions that are expressed clearly.

**Key Questions For Authors:**

Can the initial seed rubrics be created automatically in future work? How much would ORBIT’s performance vary if a different judge model was used?

**Limitations:**

yes

**Strengths And Weaknesses:**

**Strengths**

- Achieves big gains with small models, which is neat.
- Makes reward signals interpretable and avoids black-box problems.
- Incremental and filtered RL is data-efficient and stable.
- Multi-benchmark evaluation shows robustness.

**Weaknesses**

- Human-generated seed rubrics are still required.
- Judge model is fixed, which may limit robustness.
- Not totally clear how well it generalizes beyond medical-style tasks.

---

> ### Author Rebuttal · Authors · 2026-03-31
>
> We thank the reviewer for the positive feedback on the small-model gains, rubric-based rewards' interpretability, and the data efficiency of the incremental filtered RL pipeline. We understand the remaining concerns to be: (i) the need for human-authored seed rubrics, (ii) dependence on the judge model, and (iii) the scope of generalization beyond the medical setting.
> We address each point below and will revise the paper to state these boundaries more explicitly.
>
> >**Q1**: Can the initial seed rubrics be created automatically in future work?
>
> **A1:** Yes. Our new ablations suggest that partial automation is feasible, although it is not yet a drop-in replacement for expert seeding in a high-stakes setting. In the current paper, we use expert-authored seeds because medical dialogue requires reliable anchors for clinical correctness and safety during the bootstrap stage.
> Importantly, these seeds are only used to bootstrap rubric construction; ORBIT does not require dense human rubric annotation for every training example.
>
> Furthermore, to test whether seed creation can be automated, we compare expert seeds with synthetic alternatives.
> Removing retrieval grounding and using a single model to generate rubrics drops performance substantially (HealthBench-Hard: 17.89 with Human-RAG vs. 10.69 with Single-No-RAG), showing that naive automation is insufficient.
> However, on a matched 140-step subset, multi-LLM synthetic seeds reach 14.11, and synthetic seeds with contrastive filtering reach 15.60, compared with 15.34 for Human-RAG on the same subset.
> These results suggest that automatic seed construction is feasible in future work, especially with stronger generation and filtering, although it is much more expensive (at over 80 times the cost) and still benefits from expert verification in a high-stakes medical setting.
>
> | Variant | Retrieval | Seed Source         | Steps | HealthBench-Hard |
> |-|-|-|-|-|
> | Human-RAG                      | Yes       | Human seed          | 200   | 17.89            |
> | Single-No-RAG                  | No        | None                | 200   | 10.69            |
> | Human-RAG (sub)                | Yes       | Human seed          | 140   | 15.34            |
> | Multi-LLM-Syn-Direct (sub)     | No        | LLM-synthetic       | 140   | 14.11            |
> | Multi-LLM-Syn-Contrast (sub)   | No        | LLM-synthetic + contrastive filtering | 140 | 15.60 |
>
> >**Q2**: How much would ORBIT's performance vary if a different judge model was used?
>
> **A2:** We agree that using one judge family in training leaves an important robustness question.
> To address this, we ran a cross-judge study using Qwen3-4B-Instruct-2507, Qwen3-8B, Qwen3-30B-A3B-Instruct-2507, and GPT-OSS-20B as judge models, while keeping the policy model fixed as Qwen3-4B-Instruct-2507 under a 200-step training budget. All resulting policies were evaluated using a fixed GPT-OSS-120B judge for comparison.
> The experimental results show a clear monotonic trend under a shared evaluator: stronger training judges yield stronger downstream policies, and interestingly Qwen3-4B-Instruct-2507 itself as a rubrics evaluator still yields substantial performance gains.
>
> | Train Judge                 | Params | Steps | Eval Judge    | Avg Score |
> |-|-|-|-|-|
> | Qwen3-4B-Instruct-2507      | 4B     | 0     | GPT-OSS-120B  | 7.22      |
> | +Qwen3-4B-Instruct-2507     | 4B     | 200   | GPT-OSS-120B  | 13.64     |
> | +Qwen3-8B                   | 8B     | 200   | GPT-OSS-120B  | 14.56     |
> | +Qwen3-30B-A3B-Instruct-2507| 30B    | 200   | GPT-OSS-120B  | 17.89     |
> | +GPT-OSS-20B                | 20B    | 140   | GPT-OSS-120B  | 17.20     |
>
> >**Q3**: Not totally clear how well it generalizes beyond medical-style tasks.
>
> **A3:** We agree that this point should be stated more carefully.
> First, our method is designed for open-ended medical dialogue tasks.
> We aim to synthesize case-specific rubrics more effectively using simple yet efficient approaches and to investigate their issues in subsequent training scenarios.
> However, we believe that our exploration in the medical domain and the resulting performance gains can also benefit other domains.
> For example, following the reviewer’s suggestion, we have added an evaluation on an out-of-distribution open-ended medical benchmark to validate the performance of our model.
>
> | Variant              | Task Accuracy | Task Robustness | Info Coverage | Info Relevance |
> |-|-|-|-|-|
> | Qwen3-4B-Instruct-2507| 53.5           | 79.5            | 24.7          | 86.3           |
> | ORBIT-Med(2k)        | 58.0           | 78.9            | 26.2          | 87.1           |
>
> Then, for more general scenarios, we also performed some exploratory analyses by splitting InfoBench into hard and no-hard subsets, and observed certain gains.
> We will provide further details in the revised version.
>
> [1] The Dialogue That Heals: A Comprehensive Evaluation of Doctor Agents' Inquiry Capability. arXiv:2509.24958.

---

> > ### Author Rebuttal · Reviewer_55vi · 2026-04-02
> >
> > I have decided to maintain my original score, as I believe it already reflects a fair and sufficiently positive assessment of the work.

---

> > > ### Author Response · Authors · 2026-04-06
> > >
> > > Thank you very much for your kind and encouraging response. We truly appreciate your positive evaluation and thoughtful comments. We hope this work can have a positive impact on the medical domain. Thanks again  for your time and effort  in reviewing  our paper.

---

### Official Review · Reviewer_2U8d · 2026-03-16

**Soundness:** 3
**Presentation:** 3
**Significance:** 3
**Originality:** 3
**Overall Recommendation:** 4
**Confidence:** 4

**Summary:**

This paper proposes a rubric-based post-training framework for open-ended medical dialogue. The method builds case-specific rubrics, uses them to score model outputs, and then applies reinforcement learning to improve alignment on medical consultation tasks. Experiments on HealthBench-Hard and an additional instruction-following benchmark suggest that the approach can improve a small open-source model by a substantial margin, with further analysis on data scaling, filtering, and judge choices.

**Compliance With Llm Reviewing Policy:**

Affirmed.

**Final Justification:**

This paper studies an important and difficult problem, namely how to provide useful training signals for open-ended, high-stakes medical dialogue when simple scalar rewards are inadequate. The paper is generally clear, and the empirical section is substantial, with scaling studies, ablations, and judge analyses that make the practical contribution credible. My initial reservation was that the contribution appeared stronger as a carefully engineered pipeline than as a clearly separated technical idea, and I also wanted clearer validation around judge dependence, rubric quality, and robustness. The rebuttal substantially improved my assessment by adding cross-judge analyses, better component isolation, and evidence of transfer beyond the main setup. These additions made the main contribution clearer to me: not retrieval or RL alone, but the use of instance-specific, criterion-level rubric feedback as a practical supervisory interface for open-ended alignment. I still think the paper should discuss limitations more explicitly, especially around judge dependence and deployment risk, but overall the rebuttal addressed my main concerns and positively changed my evaluation.

**Key Questions For Authors:**

1. How sensitive are the results to the specific judge model and prompting setup used during training and evaluation?

2. Can the authors better isolate the contribution of the main training idea from the effects of rubric generation, retrieval design, and filtering heuristics?

3. How stable is the method across different medical dialogue domains or datasets beyond the ones studied here?

**Limitations:**

No. The paper would benefit from a more explicit discussion of judge dependence, possible brittleness of automatically generated rubrics, and the risks of using such systems in high-stakes medical settings.

**Strengths And Weaknesses:**

The paper discussed an important problem: how to train models for open-ended medical responses when simple automatic rewards are insufficient. The empirical section is fairly extensive, with multiple datasets, scaling results, ablations, and analysis of filtering and judge settings. The paper is also generally readable, and the main pipeline is easy to follow at a high level.

My main concern is that the contribution feels stronger as a carefully engineered training pipeline than as a clearly new technical idea. Many core ingredients, such as rubric-based evaluation, retrieval-augmented construction, and RL-based post-training, are already familiar, and the paper does not fully separate the gains from the central methodological contribution from those from careful system design and implementation choices. Relatedly, while the results are promising, some central components, especially rubric generation quality, judge dependence, and robustness across settings, would benefit from clearer validation before stronger conclusions are drawn.

---

> ### Author Rebuttal · Authors · 2026-03-31
>
> We thank the reviewer for the thoughtful assessment and for highlighting the need to better separate ORBIT from a strong engineering pipeline.
> Our intended contribution is not retrieval, rubric generation, or RL in isolation, but a new supervisory interface for open-ended medical dialogue: instance-specific, retrieval-grounded, criterion-level rewards that make RL feasible when a single scalar reward is inadequate.
> We add targeted evidence below on judge sensitivity, component isolation, and robustness, and we will revise the paper to state these claims and limitations more explicitly.
>
> > **Q1**: How sensitive are the results to the specific judge model and prompting setup used during training and evaluation?
>
> **A1:** We add two more experiments.
> **First**, we treat GPT-4.1 as the reference judge. We sample 200 examples from the 2k training set, using responses generated by Qwen3-30B-A3B-Instruct-2507, and evaluate the consistency of different judge models against GPT-4.1.
> From the table, we observe that a model’s reliability as a judge is strongly correlated with its underlying base capability, while GPT-OSS-120B shows particularly high concordance with GPT-4.1.
> So the rubrics-judge is judge process with noise, and we further do more analysis to evaluate if the model performance can be improved with different models.
>
> |Judge Model | Cohen's $\kappa$ $\uparrow$ | Score MAE $\downarrow$ | Spearman's $\rho$ $\uparrow$ |
> |-|-|-|-|
> |GPT-OSS-120B| **0.653**|**0.217**|**0.837**|
> |GPT-OSS-20B| 0.585|0.238|0.806|
> |Qwen3-8B|0.465|0.400|0.650|
> |Qwen3-30B-A3B-Instruct-2507|0.415|0.428|0.695|
>
> **Second**, we ran a cross-judge study in which ORBIT is trained with four judges spanning both scale and family: Qwen3-4B-Instruct-2507, Qwen3-8B, Qwen3-30B-A3B-Instruct-2507, and GPT-OSS-20B, using the same 2k medical training data and the same early 200-step budget.
> The experiments show a clear trend: stronger judges lead to stronger downstream policies under the same RL budget. Interestingly, even Qwen3-4B self-judging yields substantial improvement, suggesting the potential of a self-evolving medical LLM mechanism.
> his also suggests that our training is based on learning rubrics, rather than learning a bias from the same stronger family model.
> At the same time, we compare (chunk rubrics, case) evaluation with (single rubric, case), we find that the chunk rubrics will induce the more judge bias without training.
>
> | Train Judge|Params|Rubrics Chunk|Steps| Eval Judge|Avg Score|
> |-|-|-|-|-|-|
> | Qwen3-4B-Instruct-2507|4B| - | - | GPT-OSS-120B | 7.22 |
> | +Qwen3-4B-Instruct-2507|4B| 1 | 200   | GPT-OSS-120B |13.64 |
> | +Qwen3-8B |8B| 1 | 200   | GPT-OSS-120B |14.56 |
> | +Qwen3-30B-A3B-Instruct-2507| 30B|1| 200| GPT-OSS-120B|17.89|
> | +Qwen3-30B-A3B-Instruct-2507| 30B|4| 200| GPT-OSS-120B|14.20|
> | +GPT-OSS-20B| 20B|1| 140 | GPT-OSS-120B |17.20|
>
> >**Q2**: Can the authors better isolate the contribution of the main training idea from the effects of rubric generation, retrieval design, and filtering heuristics?
>
> **A2:** ORBIT's contribution is a rubric-based alignment pipeline for open-ended, high-stakes dialogue, where the central choice is to replace opaque scalar feedback with instance-specific, retrieval-grounded, criterion-level rewards.
> To support that point, we add a new experiment ablating the (case, rubrics) mixed retrieval part and use the single generation model to get the case-specific rubrics.
> We find the this way will induce the template rubrics such as safety without case-aware.
> And the following results further prove this thing.
>
> | Variant | Retrieval | Seed Source | Steps | HealthBench-Hard |
> |-|-|-|-|-|
> | Human-RAG| Yes | Human seed  | 200| 17.89|
> | Single-No-RAG  | No | None | 200 | 10.69|
>
> The filtering mechanism shows that pruning based on difficulty in the initial training phase retains more relevant rubrics. A detailed analysis will follow in the revised version.
>
> > **Q3**: How stable is the method across different medical dialogue domains or datasets beyond the ones studied here?
>
> **A3:**  Firstly, our scaling training data from 2k to 8k and 28k contains different sources of medical dialogue, after data-fieltering process, we train the ORBIT model and as Table 1 in our paper shows, the performance of our model improve from 27.5, 33.6 to 37.3, which can prove that our pipeline can involve different kinds of medical dialogue domains.
>
> Then, we further evaluate our ORBIT model (with 2k training samples) in a out of distribution open-ended medical benchmark [1]. Based on an evaluation over 200 cases, the following results continue to demonstrate the improved performance of our pipeline on open-ended medical tasks.
>
> |Variant|Task Accuracy|Task Robustness|Info Coverage|Info Relevance |
> |-|-|-|-|-|
> | Qwen3-4B-Instruct-2507| 53.5 | 79.5 | 24.7|86.3|
> | ORBIT-Med(2k) | 58.0 | 78.9| 26.2|87.1|
>
> [1] The Dialogue That Heals: A Comprehensive Evaluation of Doctor Agents' Inquiry Capability. arXiv:2509.24958.

---

> > ### Author Rebuttal · Reviewer_2U8d · 2026-03-31
> >
> > The author's reply has resolved my concerns, and I have updated my rating.

---

> > > ### Author Response · Authors · 2026-04-06
> > >
> > > We sincerely thank Reviewer 2U8d for the positive assessment and for increasing the score. The comments and feedback are highly valuable to us. We will carefully incorporate all of these points in the next revision. We believe these additions will make the main contributions of the paper clearer and the overall presentation more complete.

---

### Decision · Program_Chairs · 2026-04-30

**Decision:**

Accept (regular)

**Comment:**

The paper proposes a rubric-based reinforcement learning approach for open-ended domains called ORBIT. The idea is to replace fixed reward models with dynamically generated evaluation rubrics used to guide training via LLM-based feedback and results in improvements in performance with minimal data.
Reviews are consistent and highlight the importance of the problem, clarity of presentation, and thorough empirical evaluation, with all recommending weak accept after rebuttal. They also raise concerns, such as the reliance on human seed rubrics, sensitivity to judge models, and limited validation of robustness, safety, and generalization; most of which were addressed during the rebuttals with additional experiments.